# First steps into the cloud: Using Amazon data storage and computing with Python notebooks

**Daniel J. Pollak**[1,2], **Gautam Chawla**[1,2], **Andrey Andreev**[1,2]*, **David A. Prober**[1,2]

**1** Division of Biology and Biological Engineering, California Institute of Technology, Pasadena, California, United States of America, **2** Tianqiao and Chrissy Chen Institute for Neuroscience, California Institute of Technology, Pasadena, California, United States of America

* aandreev@caltech.edu

## Abstract

With the oncoming age of big data, biologists are encountering more use cases for cloud-based computing to streamline data processing and storage. Unfortunately, cloud platforms are difficult to learn, and there are few resources for biologists to demystify them. We have developed a guide for experimental biologists to set up cloud processing on Amazon Web Services to cheaply outsource data processing and storage. Here we provide a guide for setting up a computing environment in the cloud and showcase examples of using Python and Julia programming languages. We present example calcium imaging data in the zebra-fish brain and corresponding analysis using suite2p software. Tools for budget and user management are further discussed in the attached protocol. Using this guide, researchers with limited coding experience can get started with cloud-based computing or move existing coding infrastructure into the cloud environment.

## Introduction

Modern life sciences require immense technical knowledge to process data and complicated protocols to ensure reproducibility. For a variety of reasons, researchers may want to migrate on-premises data processing (i.e., workstations and personal computers) to centralized computing systems, including "cloud" computing environments. However, migration is complex and requires specialized knowledge. The ecosystem of cloud storage/computing services is expansive enough to overwhelm newcomers from academia.

Moving data processing and management to the cloud can have several advantages for biologists. First, cloud platforms do not require purchasing of data storage resources upfront. Users only pay for space as needed, unlike physical workstations, when a whole block of spaces is bought at once and then slowly filled. Many biological experiments, such as microscopy experiments, generate large volumes of data so quickly that personal computers become insufficient for storing data, much less accomplishing intensive data processing tasks [1]. Cloud processing can solve this issue by allowing gradual expansion of storage and processing

**Data Availability Statement:** Supporting code and calcium imaging data is available from the Caltech Data repository (doi.org/10.22002/6ejqf-qm267).

**Funding:** The authors report the following sources of funding: NIH (R35 NS122172) awarded to DAP,

NIH (T32 NS105595) awarded to DJP, and Caltech/
Amazon AI4Science Cloud Credits Program grant
awarded to DAP.

**Competing interests:** The authors have declared
that no competing interests exist.

without up-front costs. One increasingly popular option for cloud processing is Amazon Web Services (AWS), but the barriers to entry are high for those who do not know where to begin.

## Cloud infrastructure

Currently, several cloud computing platforms exist, e.g. those provided by Amazon (AWS), Google (Colab), and Microsoft (Azure). The process for setting up a workflow differs significantly across cloud providers and requires platform-specific knowledge. Amazon collaborates with our home institution (Caltech) and provides research credits to support computing, and also allows a wider range of services and tools than other providers. Thus, we focus on AWS in this guide.

1. Infrastructure elements
   Main three elements of AWS infrastructure that we consider here are: computing resources, data storage resources, and billing and payment process.
   Currently users can choose from 200+ AWS services, but two services form the foundation of computing and storage: Simple Storage Service (S3) and Elastic Computing 2 (EC2). S3 serves as a long-term storage within AWS, and EC2 provides a pay-by-the-second service to rent computing resources ("virtual machines"; VMs) to analyze data stored in S3. When working in a Python-based environment, EC2 connects to S3 using AWS's boto3 library (Fig 1). A protocol for setup and use of these services is published on dx.doi.org/10.17504/ protocols.io.rm7vz3z4xgx1/v1 (see S1 File). The protocol on protocols.io demonstrates how to set up an AWS organization, how to load data to S3, how to set up computing instances in EC2, and finally how to configure and run the analyses we show here.

2. Organizing collaboration / lab management
   It is important for new AWS users to understand payment structure and methods that are used. To facilitate productive and cost-effective AWS usage, allocating access to users and managing budget is paramount. The primary tool for accomplishing these tasks is the AWS Organizations framework, where existing users are added to the organization, new accounts

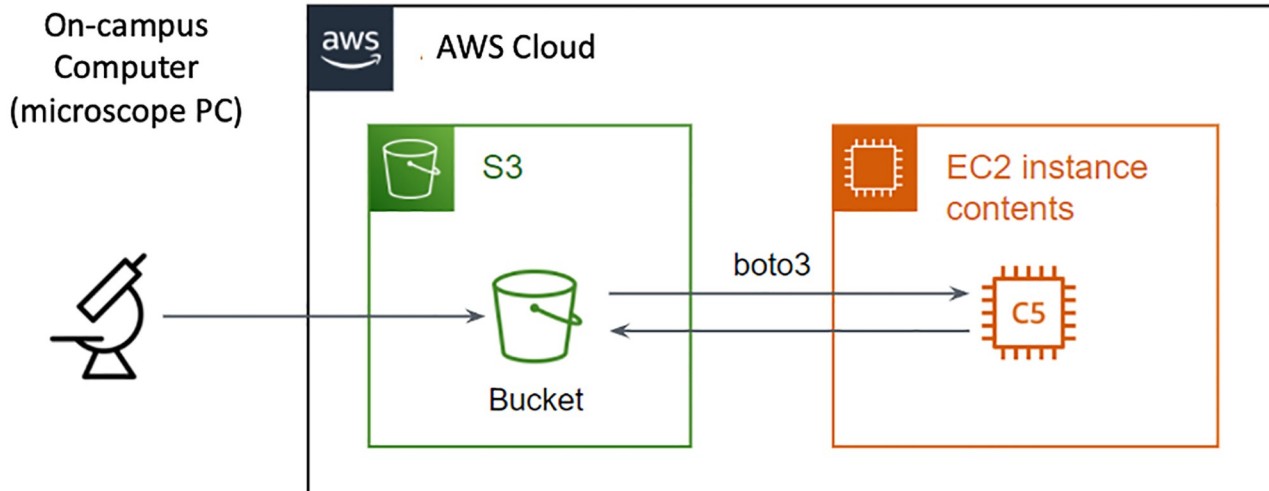

**Fig 1. Outline of the pipeline used to work with AWS.** Data from the microscope is moved to S3 data storage using university-provided networks. Data transfer rate can range between 10-100Mb/s. Within AWS cloud infrastructure, data is moved at >10GB/s from storage to EC2 virtual machines to be processed through a Python or Julia Jupyter notebook.

are created, and billing is centralized. This framework gives lab members access to AWS services using their personal Amazon accounts, but without using their personal finances.

By default, every AWS user account requires a credit card for billing. The Organizations framework transfers the burden of payment away from individual users. Payments are made using a credit card or Research Credits (Fig 2). Research credits are through Amazon-managed program together with Caltech and other universities, as well as independently by Amazon. Credits that can be redeemed from AWS and are applied to the whole Organization account. The account administrator can invite new users with existing AWS accounts or create new accounts associated with the Organization from the beginning. We recommend the latter route to avoid personal liability so that in the worst-case scenario, the Organization will face cost overruns, not individual researchers.

Whereas traditional business models charge an amount agreed upon in a quote before the transaction is completed, cloud computing providers use the pay-as-you-go model of billing. Academics usually use the former business model, but in the case of cloud-computing, customers cannot precisely predict the eventual cost. Analyses often need to be re-run, evaluated, and fine-tuned. Each of these steps involves a billable service for seconds of computing, for seconds of storing gigabytes of data, and often for gigabytes of data moved through the internet from the user's computer to the cloud provider. Without an understanding of this billing regime, unexpected costs will arise.

To avoid unexpected costs, EC2 users need to keep in mind the difference between closing a browser window (for example, ending a Jupyter session), shutting down an instance, and terminating an instance. When a user begins an AWS session, they navigate to their dashboard (Fig 3) and start an existing instance, at which point AWS starts billing by the second. AWS beginners are prone to making the mistake of closing their AWS dashboard window, assuming that will end their session. However, AWS does not stop billing until an instance is manually shut down, and this mistake can incur serious costs. Another common error is confusing shutting down an instance, which simply shuts down the EC2 instance operating system (similar to turning a personal computer off), with terminating an instance, which destroys that instance, possibly resulting in data loss.

The AWS Budgets tool controls costs and associated alerts. An Organization's administrator can set limits on spending (e.g. $100 per month across all services) and associate alerts or actions on a per-account basis. By default, budgets are set for the whole organization, so to limit spending by an individual account, the administrator needs to add a user-specific filter. An administrator (or any other user) can be alerted if a limited budget has been exceeded. The account can then be restricted from performing certain actions such as starting new EC2 instances or accessing any other AWS services.

## From theory to practice: How new users can expect to interact and benefit from cloud computing

Using AWS computing allows a researcher to transfer existing code to the cloud and run it with better computing resources. This can usually be accomplished without major changes to the code. Here we present a guide for using Python and Julia [2, 3] to analyze data using AWS. We show how to install software packages on this cloud platform, how to deposit and access data with a distributed storage service, and how to leverage these resources to successfully navigate the ever-strengthening deluge of big data. We aim to facilitate robust and reproducible collaboration with commercial cloud computing. After adopting these tools, we believe many research groups will benefit from increased bandwidth of existing pipelines and making

(A)

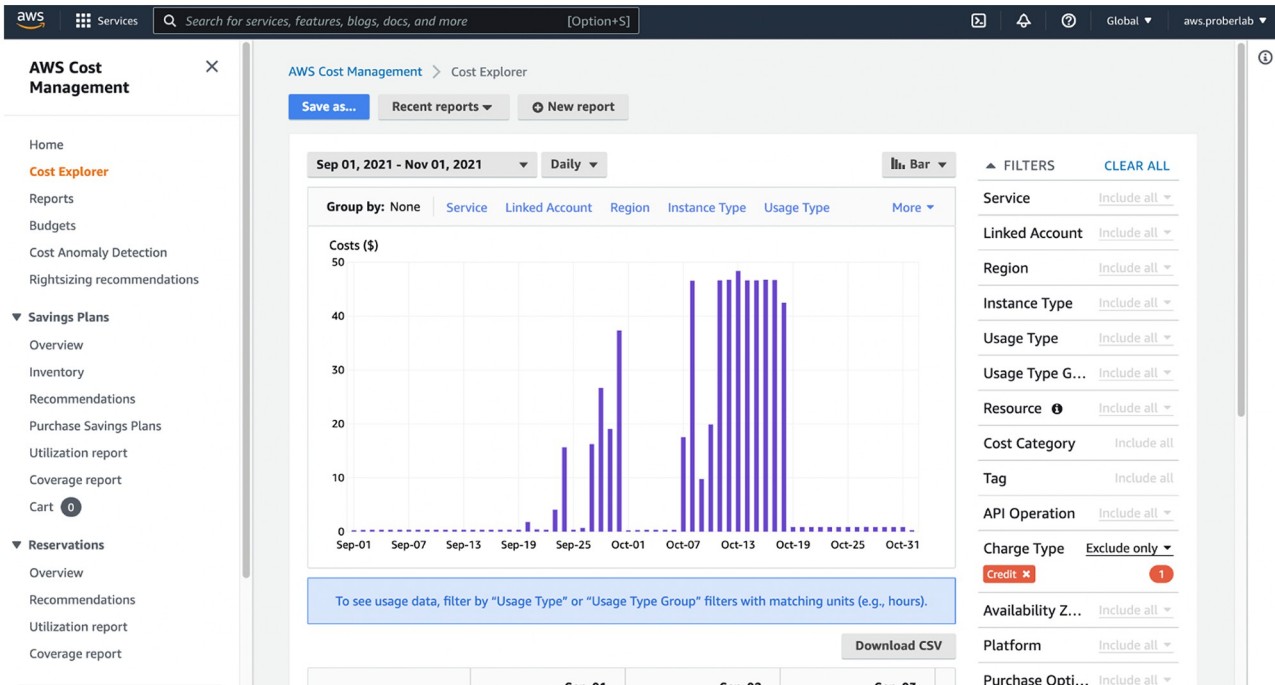

(B)

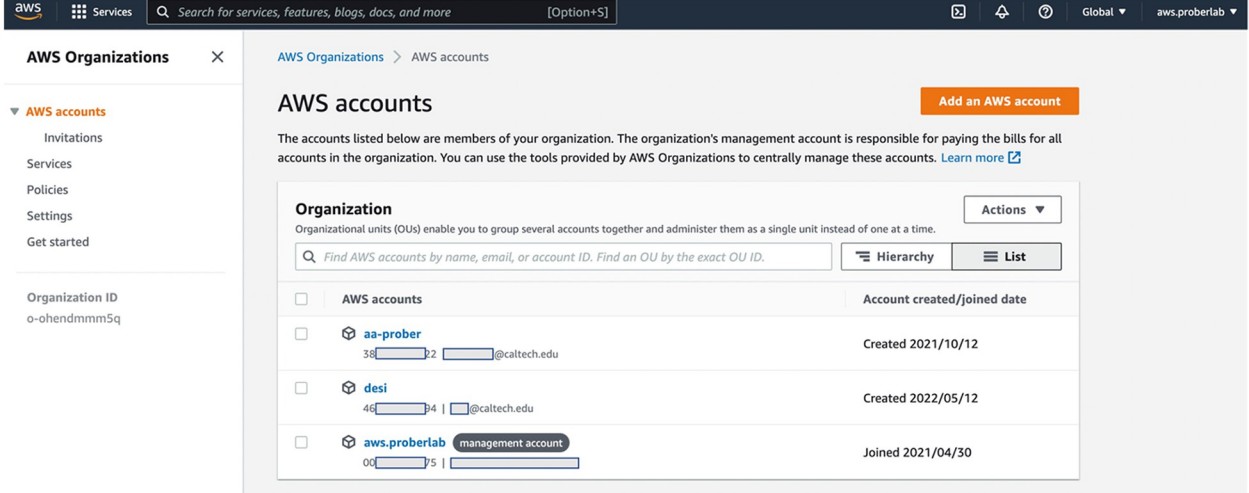

**Fig 2.** (Top) AWS Organizations portal lists all users and allows addition and creation of new users within the organization. Responsibility for billing of these users are fully on the organization. Creation of new users through Organizations interface is recommended because it allows easy removal of accounts after, for example, student leaves, or the project is finished. (Bottom) Billing Dashboard provides access to multiple tools to manage spending on the level of organization. One of the tools for control is the AWS Cost Management Cost Explorer console. Administrator can check spending by service type (for example, only computing or only storage) and by user account.

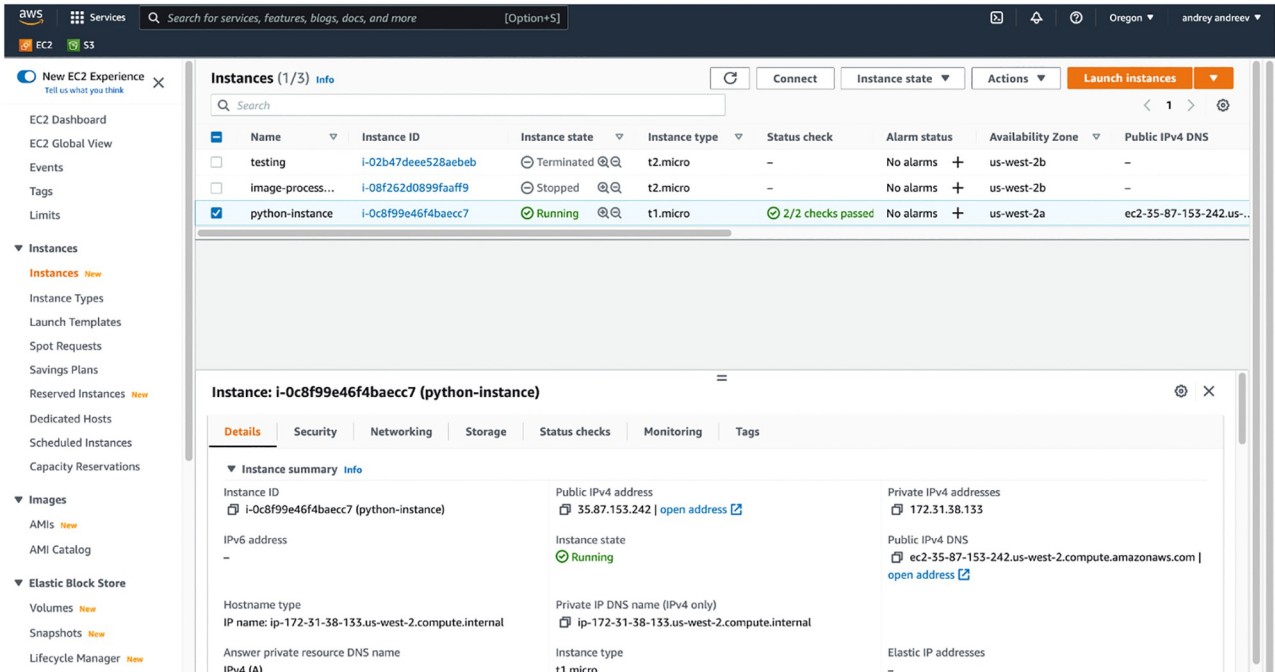

**Fig 3. EC2 dashboard showing all launched, stopped, and recently terminated instances.** Running instances are virtual machines, and you are being billed by the second whether you are connected to them or not. Stopped instances are billed for space required to store their memory and data. Stopped instances can be re-started, but state of the memory (RAM) is not guaranteed to be preserved.

computationally and spatially intensive tools such as suite2p [4] tractable for researchers who do not have access to sufficient computing resources locally.

1. Setting up a new instance

   Work with AWS EC2 starts with launching an instance, a VM with specific hardware and software. AWS provides a long list of possible instance types with certain memory capacities (RAM), CPU capacities, and specialized GPUs (Fig 4). Users can also specify operating system, including Linux or Windows. A crucial step in launching an EC2 instance is selecting appropriate access rights, which determines how the user can connect to the instance. Usually, connection to an instance is made *via* web-interface or using a terminal from another computer (using *ssh* protocol), so appropriate access rights need to be selected before starting the instance.

2. Using Jupyter notebooks for processing

   There are several ways to run python code on EC2 instances. Using Jupyter notebooks might be the easiest as it provides an interface for generating plots and displaying images, but it requires addressing security issues. Jupyter notebooks run on an EC2 instance, but users connect to it via a separate web browser window. To comply with firewall constraints, the user specifies that Jupyter will use port 8000.

3. Reproducibility

   At some point in the research cycle, a research team will find it necessary to share their analysis methods. Whereas open-sourcing source code is useful, it is often not particularly workable because the myriad installations and configurations required to get the code to work on any one machine are difficult to reproduce blindly.

(A)

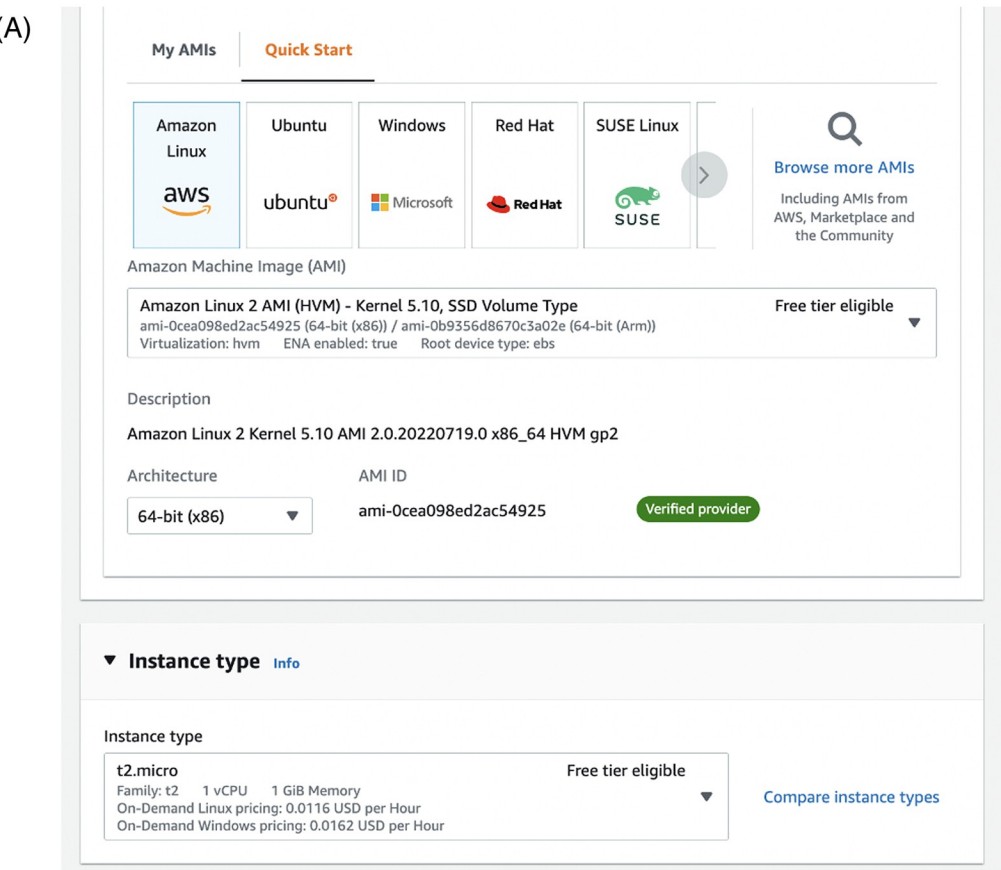

(B)

**Fig 4.** (Right) To launch an EC2 instance, you need to pick an operating system (standard Amazon Linux will provide flexibility for new users) and technical parameters, such as CPU power and memory (RAM). Standard unit of computing power is virtual CPU, vCPU. There is no linear relationship between physical processors and vCPU, but one virtual CPU unit corresponds to a single thread. Correctly selecting the number of vCPUs for your code might require experimenting. Some instances come with graphical processing units (GPU), other instances provide access to fast solid-state drive storage. (Right) Second part of launching an instance is selecting access rules. It includes setting up private-public key pair for passwordless login, and allowing firewall rules for SSH traffic (terminal) and potentially also traffic on ports associated with Jupyter notebook server (such as 8000 port).

Cloud computing allows users to share access to the exact VM image that they used to get their analyses working, all the way down to the python dependencies installed. It is beyond the scope of this paper to enumerate the steps for this, rather we intend to aid new users in getting acquainted with AWS services. However, we believe that sharing reproducible machine images will prove to be quite ubiquitous in the future, as it is the only method that can overcome computational limitations to reproducibility.

Working with virtual machines allows creation of "Images" or snapshots of the VM that contains all the installed software, dependencies, and data, allowing for researchers to reproduce each others' analyses with far more ease. The Image can be shared with anyone, and by design AWS guarantees that the analysis pipeline will function identically for each person that runs it. Python environments frequently "break", meaning some code dependencies interfere with others such that the environment becomes unusable, requiring users to reinstall everything. VMs circumvent this "broken environment" issue because a saved snapshot of a properly functioning VM image allows users to seamlessly roll back to a functioning state.

## Expected results

To help users enter cloud computing practice, an example dataset and code to process it is provided as supplemental data. Neural activity data presented here is collected using a custom two-photon light-sheet microscope based on previous publication [5]. Total laser power of 300mW was used at 920nm wavelength to collect spontaneous neural activity data from larval zebrafish expressing pan-neuronal nuclear-localized calcium indicator GCaMP6s. Animals were used for imaging at 5 and 6 days post fertilization and euthanized immediately after imaging experiments in tricaine solution.

By configuring a virtual machine on AWS and uploading data to S3 (see Fig 1), users can transfer data processing and analysis from on-premises workstations to cloud environment. To work with our data we first transferred a large set of imaging TIF files (see associated content hosted on CaltechDATA, File S5 in S1 Data) using Suite2P (see associated content hosted on CaltechDATA, File S2 in S1 Data). Then, K-means clustering was used to identify functionally correlated regions of interest (Fig 5).

Alongside this protocol, we present several Python and Julia notebooks that showcase work with zebrafish brain calcium activity data collected using a custom two-photon light-sheet microscope. To apply these notebooks, data has to be deposited in an S3 bucket, and then File S2 in S1 Data, *S2_run_suite2p_aws.ipynb* will download data and perform single-cell segmentation using Suite2P. The result data of neural activity traces and locations will be saved locally on the computing instance's disk.

To quickly get an understanding of data quality and start getting an intuition for large-scale patterns in imaging data, we have found it useful to produce animated *gif* files with projections of the entire recording. We generate such gifs of maximum intensity projection and standard deviation projection in time using Julia (File S3 in S1 Data, notebook *S3_exploratory_gifs_Julia.ipynb*). Next, we applied processing that reveals functional units of correlated cells, namely K-means clustering analysis. Code to produce visualization presented here can be found in associated content hosted on CaltechDATA, File S4 in S1 Data.

## Conclusion

At the outset of a project, it can be impossible to predict the computational requirements for the full set of data analysis tasks that an investigator will want to accomplish. The computational scope of analyses required to understand data can easily exceed the capabilities of a single workstation, and will require a more powerful computer, for example when memory or

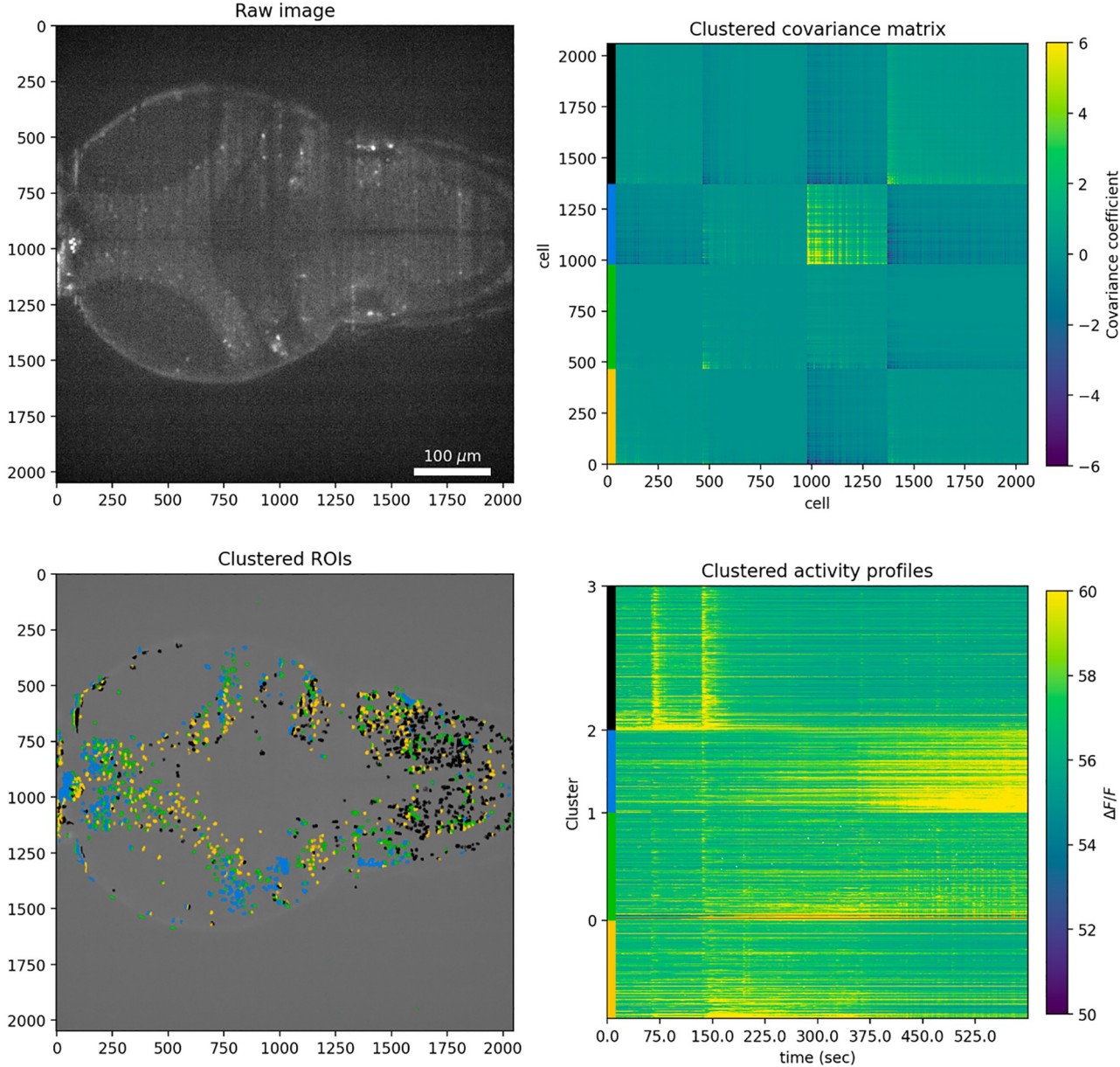

**Fig 5. Analysis pipeline for Suite2P-processed data with k-means clustering for k = 4 (four clusters).** Top left: first raw image of the timeseries imaging data. Top right: k-means-clustered covariance matrix of ROIs activity. Color bars on the left correspond to different clusters. Bottom left: Spatial distribution of Suite2P neuron ROIs colored according to covariance matrix clustering. Colors indicate cluster membership according to k-means clustering of the covariance matrix. Bottom right: Activity traces for each Suite2P ROI. Color bars on the left indicate cluster membership according to k-means clustering of the covariance matrix.

GPU-based computing are limiting factors. Cloud computing provides profound flexibility in computational capacity, allowing researchers to pilot resource-intensive tasks with minimal cost. With cloud computing, a scientist can test their code on a variety of configurations and rent a more powerful computer when computing needs evolve. This flexibility requires additional attention to billing, but this consideration is tractable, and allows scientists to dramatically accelerate the research process.

## Supporting information

**S1 File. Step-by-step protocol to set up AWS organization and run Python code, also available on Protocols.io.**
(PDF)

**S1 Data.**
(DOCX)

## Acknowledgments

The authors gratefully acknowledge Tom Morrell and Dr. Kristin Briney for support. Many thanks to Justin Bois for instructing course BE/Bi 103b.

### Associated content

Protocols.io link: dx.doi.org/10.17504/protocols.io.rm7vz3z4xgx1/v1.

### Ethics declarations

Experiments involving zebrafish were performed according to the California Institute of Technology Institutional Animal Care and Use Committee (IACUC) guidelines and by the Office of Laboratory Animal Resources at the California Institute of Technology.

## Author Contributions

**Conceptualization:** Andrey Andreev.

**Software:** Daniel J. Pollak, Gautam Chawla, Andrey Andreev.

**Supervision:** David A. Prober.

**Writing – original draft:** Daniel J. Pollak, Andrey Andreev.

**Writing – review & editing:** Daniel J. Pollak, Andrey Andreev.

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
