## [Decision Letter · Decision Letter 0]

8 Feb 2022

PONE-D-21-39110First Steps into the Cloud Using Amazon Data Storage and Computing with Python NotebooksPLOS ONE

Dear Dr. Andreev,

Thank you for submitting your manuscript to PLOS ONE. After careful consideration, we feel that it has merit but does not fully meet PLOS ONE’s publication criteria as it currently stands. Therefore, we invite you to submit a revised version of the manuscript that addresses the points raised during the review process.

We look forward to receiving your revised manuscript.

Kind regards,

Chee Kong Chui, PhD

Academic Editor

PLOS ONE

Journal Requirements:

Additional Editor Comments:

The manuscript too brief to be qualified as a journal paper but I like the topic. Cloud computing has practical importance. The author should do a major revision of the paper. The paper should present sufficient information so that a reader can reproduce the work, and also provide sufficient insights so that a reader can appreciate the problems and contributions.

Reviewers' comments:

Reviewer's Responses to Questions

**Comments to the Author**

1. Does the manuscript report a protocol which is of utility to the research community and adds value to the published literature?

Reviewer #1: Yes

2. Has the protocol been described in sufficient detail?

Descriptions of methods and reagents contained in the step-by-step protocol should be reported in sufficient detail for another researcher to reproduce all experiments and analyses. The protocol should describe the appropriate controls, sample sizes and replication needed to ensure that the data are robust and reproducible.

Reviewer #1: Partly

3. Does the protocol describe a validated method?

Reviewer #1: Yes

4. If the manuscript contains new data, have the authors made this data fully available?

Reviewer #1: N/A

**5. Is the article presented in an intelligible fashion and written in standard English?**

Reviewer #1: Yes

6. Review Comments to the Author

Reviewer #1: The article presents a protocol on using Amazon Web Services and computing with Python notebooks. It also comes with a step-by-step guide attached as a supporting file. The guide is comprehensive and suitable for users with limited programming experience. Programming languages covered include Python and Julia. Overall, the article is fine but requires some revision as per my comments below:

The author did a brief walkthrough using data from their work regarding zebrafish brain calcium activity. Although it is understood that such articles are meant to be brief, the current article is too brief. It is good that the supporting file provides step-by-step guide with screenshots to guide users. But the main article should have substantial amount of information and description to allow users to follow through and understand each main step of analysis with reference to the validated case (of zebrafish data). In the article’s current state, a person into his/her “first steps” will not understand well. Please improve on this.

In this article and/or the supporting file, it will be good to also include some potential issues that the user might face while following through the steps, as well as troubleshooting methods.

As this is a protocol for “First steps into the cloud” as described by the title, the author should address the concerns that a beginner might have regarding cloud computing. For example, although the protocol is focused on Amazon, you should also compare AWS to other IaaS providers. What are the advantages and disadvantages? Is it easier to use? Is it more cost effective? Can the described methods in this article be applied directly to other IaaS in the same manner?

3rd paragraph of Introduction: “Replicability challenges faced by academics are formidable”. The word “formidable” is unclear and confusing. Is it in a good way, or bad way?

3rd paragraph of sub-heading Introduction: “As the biological sciences become more computationally- and data-driven,collaborators and information consumers increasingly try to replicate both hardware and software”. The word “replicate” is not a good choice of word here. Replicate suggests the need to make exact copies of hardware and software, which may not be necessary. Please rephrase.

1st paragraph of sub-heading Materials and methods: “AWS uses a framework called Organizations to control who can use AWS resources outside of the Organization administrator”. Please use academic writing. It sounds very much like the author is casually speaking.

Similarly, in the same paragraph, the author wrote “Which AWS services will we be using?” before answering the question. Having ideas/points written in the form of questions is not appropriate for a proper article. Please change it.

In fact, there are other instances where the English expressions used are too casual. Please check through the entire article and change to academic writing style.

Last bullet-point of sub-heading Costs and use cases: “For a primer on best practices for data storage, see [1].” Please include the authors’ names every time you include a reference. This is not only for better academic writing, but also allows readers to know the work immediately without going around to find that reference.

For sub-heading Costs and use cases: Costs and use cases seem to be very different topics. Unless the author is trying to relate costs to use cases (which I do not observe), it should be kept as separate sub-headings.

7. PLOS authors have the option to publish the peer review history of their article (what does this mean?). If published, this will include your full peer review and any attached files.

Reviewer #1: No

---

## [Author Response · Author response to Decision Letter 0]

6 Oct 2022

Dear PLOS One editor,

We are thankful for the opportunity to present our work and communicate with larger research community about practical applications and challenges of cloud computing on commercial platforms such as Amazon Web Services.

As suggested by the editor and the reviewer, we have performed major revision of the paper to provide more details and insights for the readers even not familiar with cloud computing. Specifically, we expanded discussion of every section: from concerns about budgeting and billing, to description of process to start virtual machines.

Below we provide details response to reviewer’s comments.

Additional Editor Comments:

The manuscript too brief to be qualified as a journal paper but I like the topic. Cloud computing has practical importance. The author should do a major revision of the paper. The paper should present sufficient information so that a reader can reproduce the work, and also provide sufficient insights so that a reader can appreciate the problems and contributions.

We have expanded text to include more details in every section. Cloud infrastructure is discussed in more details to help those new to the concept of distributed computing, and specifically to Amazon’s cloud architecture. We have expanded section about organization of cloud computing for the whole lab, discussing how users can have separate budgets and access rights. We focused on how researchers can avoid accidentally wasting money. We have also included a step-by-step guide on setting up computing environment, so that just reading the paper (even before opening the protocol.io document) users can get valuable insights and reference material. We have also included more screenshots. We have discussed how python language and Jupyter notebooks can be leveraged to increase reproducibility of the research.

6. Review Comments to the Author

Although it is understood that such articles are meant to be brief, the current article is too brief. […] But the main article should have substantial amount of information and description to allow users to follow through and understand each main step of analysis with reference to the validated case (of zebrafish data). In the article’s current state, a person into his/her “first steps” will not understand well. Please improve on this.

We have expanded the amount of text covering motivation and explanation for each section in the manuscript. We provided more detailed discussion of concepts used in cloud computing and Amazon cloud, specifically.

In this article and/or the supporting file, it will be good to also include some potential issues that the user might face while following through the steps, as well as troubleshooting methods.

We have highlighted several issues that users can face: inability to connect to new instance (that can be fixed by firewall settings – we provide information) and budget over-runs (which can be alleviated by managing budgets and users – we provide discussion of some tools for that)

As this is a protocol for “First steps into the cloud” as described by the title, the author should address the concerns that a beginner might have regarding cloud computing. For example, although the protocol is focused on Amazon, you should also compare AWS to other IaaS providers. […]

We have briefly compared Amazon with other cloud providers; however, we note that all providers are different from each other, and translating knowledge between them is not easy. Hence, here we explicitly focus on Amazon provider as they provide the largest suite of tools, and we have the most experience with Amazon. 

3rd paragraph of Introduction: “Replicability challenges faced by academics are formidable”. The word “formidable” is unclear and confusing. Is it in a good way, or bad way?

3rd paragraph of sub-heading Introduction: “As the biological sciences become more computationally- and data-driven,collaborators and information consumers increasingly try to replicate both hardware and software”. The word “replicate” is not a good choice of word here. Replicate suggests the need to make exact copies of hardware and software, which may not be necessary. Please rephrase.

Edited language

1st paragraph of sub-heading Materials and methods: “AWS uses a framework called Organizations to control who can use AWS resources outside of the Organization administrator”. Please use academic writing. It sounds very much like the author is casually speaking.

Similarly, in the same paragraph, the author wrote “Which AWS services will we be using?” before answering the question. Having ideas/points written in the form of questions is not appropriate for a proper article. Please change it.

In fact, there are other instances where the English expressions used are too casual. Please check through the entire article and change to academic writing style.

Significantly changed language to be more within academic style, and more precise

Last bullet-point of sub-heading Costs and use cases: “For a primer on best practices for data storage, see [1].” Please include the authors’ names every time you include a reference. This is not only for better academic writing, but also allows readers to know the work immediately without going around to find that reference.

Changed references to PLOS style 

For sub-heading Costs and use cases: Costs and use cases seem to be very different topics. Unless the author is trying to relate costs to use cases (which I do not observe), it should be kept as separate sub-headings.

Edited to split costs and applications in separate sections

---

## [Decision Letter · Decision Letter 1]

25 Oct 2022

PONE-D-21-39110R1First Steps into the Cloud Using Amazon Data Storage and Computing with Python NotebooksPLOS ONE

Dear Dr. Andreev,

Thank you for submitting your manuscript to PLOS ONE. After careful consideration, we feel that it has merit but does not fully meet PLOS ONE’s publication criteria as it currently stands. Therefore, we invite you to submit a revised version of the manuscript that addresses the points raised during the review process. Kindly revise your manuscript as per reviewers comments. 

We look forward to receiving your revised manuscript.

Kind regards,

Mudassir Khan, Ph.D

Academic Editor

PLOS ONE

Journal Requirements:

Reviewers' comments:

Reviewer's Responses to Questions

**Comments to the Author**

1. Does the manuscript report a protocol which is of utility to the research community and adds value to the published literature?

Reviewer #1: Yes

Reviewer #2: Yes

2. Has the protocol been described in sufficient detail?

To answer this question, please click the link to protocols.io in the Materials and Methods section of the manuscript (if a link has been provided) or consult the step-by-step protocol in the Supporting Information files.

The step-by-step protocol should contain sufficient detail for another researcher to be able to reproduce all experiments and analyses.

Reviewer #1: Yes

Reviewer #2: Yes

3. Does the protocol describe a validated method?

Reviewer #1: Yes

Reviewer #2: Yes

4. If the manuscript contains new data, have the authors made this data fully available?

Reviewer #1: N/A

Reviewer #2: N/A

**5. Is the article presented in an intelligible fashion and written in standard English?**

Reviewer #1: Yes

Reviewer #2: Yes

6. Review Comments to the Author

Reviewer #1: Authors have amended the article according to the previous comments made. Therefore, the article is better now.

Reviewer #2: Dear authors

The screenshots you presented in this paper is not clear and create some confusion. Try to avoid use of screenshots.

7. PLOS authors have the option to publish the peer review history of their article (what does this mean?). If published, this will include your full peer review and any attached files.

Reviewer #1: No

Reviewer #2: No

---

## [Author Response · Author response to Decision Letter 1]

4 Nov 2022

To the editor,

We thank the editor and the reviewers for feedback and timely response to our re-submission.

We have amended the manuscript by adding DOI links to the data (hosted on CaltechDATA archive) and Protocols.IO.

We would like to specifically address comment from the Reviewer #2:

The screenshots you presented in this paper is not clear and create some confusion. Try to avoid use of screenshots.

We feel that using screenshots in the main article text helps us to “demystify” cloud computing interfaces, which is the main goal of the article. We reviewed the captions to make sure they are as clear as possible and contain enough information to guide the reader.

---

## [Editor Report · Decision Letter 2]

15 Nov 2022

First Steps into the Cloud Using Amazon Data Storage and Computing with Python Notebooks

PONE-D-21-39110R2

Dear Andrey Andreev,

We’re pleased to inform you that your manuscript has been judged scientifically suitable for publication and will be formally accepted for publication once it meets all outstanding technical requirements.

Kind regards,

Mudassir Khan, Ph.D

Academic Editor

PLOS ONE

Additional Editor Comments (optional):

Thanks for submitting your article in PLOS ONE. I hope you understand the journal reviewing policies. Thanks for your cooperation and understanding.
---

## [Editor Report · Acceptance letter]

22 Nov 2022

PONE-D-21-39110R2 

First steps into the Cloud: Using Amazon data storage and computing with Python notebooks 

Dear Dr. Andreev:

I'm pleased to inform you that your manuscript has been deemed suitable for publication in PLOS ONE. Congratulations! Your manuscript is now with our production department. 

Kind regards, 

on behalf of

Dr. Mudassir Khan 

Academic Editor

PLOS ONE